# A role for the cholinergic neuron circadian clock in RNA metabolism and mediating neurodegeneration

Sharon B Tam[1], Nathan J Waldeck[2], Matthew Wright[1,3], Jelena Mojsilovic-Petrovic[1], Erin M Baker[1], Evangelos Kiskinis[1,3], Joseph Bass[2], Robert G Kalb[1]

**Circadian clocks are encoded by a transcription-translation feedback loop that aligns physiological processes with the solar cycle. Previous work linking the circadian clock to the regulation of RNA-binding proteins (RBPs) provides a foundation for the vital examination of their mechanistic connections in the context of amyotrophic lateral sclerosis (ALS)—a fatal neurodegenerative disease commonly marked by disrupted RBP function. Here, we reveal that the spinal cord cholinergic neuron rhythmic transcriptome is enriched for genes associated with ALS and other neurodegenerative diseases. We show that there is time-of-day-dependent expression of ALS-linked RBP transcripts and rhythmic alternative splicing of genes involved in microtubule cytoskeleton organization, intracellular trafficking, and synaptic function. Through in silico analysis of RNA sequencing data from sporadic ALS patients, we find that gene expression profiles altered in disease correspond with rhythmic gene networks. Finally, we report that clock disruption through cholinergic neuron-specific deletion of clock activator BMAL1 increases neurodegeneration and drives time-of-day-dependent alternative splicing of RNA processing genes. Our results establish a role for the cholinergic neuron circadian clock in RNA metabolism and mediating neurodegeneration.**

## Introduction

The circadian system acts as a biological timekeeper that aligns physiological energetic cycles with the day/night cycle. In mammals, the molecular clock is encoded by a transcription–translation feedback loop comprised of activators (CLOCK and BMAL1) that induce the transcription of repressors (PERs and CRYs), which feedback to inhibit the forward limb in a cycle that repeats itself every ~24 h, with a stabilizing loop consisting of REV-ERB and ROR transcription factors that modulate BMAL1 expression (Bass, 2012). In recent decades, clock transcriptional regulation has been characterized in diverse mammalian tissues and is estimated to drive the rhythmic expression of 10–15% of all mRNAs within a given tissue (Panda et al, 2002; Lowrey & Takahashi, 2004; Vollmers et al, 2009). Intriguingly, recent studies in liver have demonstrated that most of the genes that exhibit rhythmic mRNA expression do not exhibit rhythmic expression of the corresponding intron-containing pre-mRNA, suggesting that post-transcriptional modifications play a major role in the regulation of circadian gene expression (Kojima et al, 2011; Koike et al, 2012; Menet et al, 2012; Green, 2018).

Emerging evidence points to a connection between the circadian clock and RNA processing events, such as methylation, poly-adenylation, and alternative splicing (Kojima et al, 2012; Fustin et al, 2013; Marcheva et al, 2020). Time-of-day-dependent production of alternatively spliced transcripts has been previously reported in plants, flies, and mice, primarily in neuronal cell-types (Sanchez et al, 2010; Hughes et al, 2012; McGlincy et al, 2012; Wang et al, 2018; Marcheva et al, 2020; Yang et al, 2020). Alternative splicing allows a single gene to produce multiple distinct mRNA transcripts in a tissue- and cell-type specific manner, which contributes to transcriptomic diversity and enables cells to perform complex and specialized functions (Young et al, 1981; Chen & Manley, 2009). This process is strongly regulated by RNA-binding proteins (RBPs), which bind to specific regulatory sequence motifs in pre-mRNA and either enhance or suppress spliceosome recruitment to splice sites (Ule & Blencowe, 2019). Although RBPs are ubiquitously expressed, cells and tissue within the central nervous system (CNS) demonstrate greater disease vulnerability to RBP dysregulation due to their higher frequency of alternative splicing (Yeo et al, 2004; Pan et al, 2008).

Disrupted RBP homeostasis is a hallmark of many neurological diseases, including amyotrophic lateral sclerosis (ALS), which is a progressive neurodegenerative disease that originates in upper and lower motor neurons in the brain and spinal cord (Nussbacher et al, 2019; Prashad & Gopal, 2021). Genetic studies have revealed associations between ALS and mutations in genes encoding a number of RBPs, including TDP-43, FUS, ATXN2, TAF15, EWSR1,

[1]The Ken and Ruth Davee Department of Neurology, Northwestern University Feinberg School of Medicine, Chicago, IL, USA [2]Department of Medicine, Division of Endocrinology, Metabolism, and Molecular Medicine, Northwestern University Feinberg School of Medicine, Chicago, IL, USA [3]Department of Neuroscience, Northwestern University Feinberg School of Medicine, Chicago, IL, USA

Correspondence: robert.kalb1@northwestern.edu

hnRNPA1, hnRNPA2B1, MATR3, and TIA1 (Kapeli et al, 2017; Zhao et al, 2018). These RBPs are key regulators of RNA metabolism and neuronal health, and ALS-linked mutations have been shown to vastly disrupt RNA processes, such as alternative splicing, mRNA transport, liquid phase transitions, and local translation (Kapeli et al, 2017; Nagano & Araki, 2021). For example, many of the proteins that copurify with TDP-43 are associated with RNA splicing and translation (Freibaum et al, 2010), and ALS-linked mutations in TDP-43 were shown to impair mRNA axonal transport and alter the fluid properties of TDP-43–positive liquid droplets (Alami et al, 2014; Schmidt & Rohatgi, 2016).

Previous work linking the circadian clock to RBP function provides a foundation for the critical examination of their mechanistic connections in the context of ALS (Wang et al, 2018; Marcheva et al, 2020). The role of the lower motor neuron (LMN) clock in disrupted RBP function and alternative splicing as it relates to ALS disease pathophysiology is not known. Here, we perform transcriptomic analysis of spinal cord cholinergic neurons to elucidate the time-of-day-dependent effects of cell-type-specific clock disruption. We report that the cholinergic neuron rhythmic transcriptome is enriched for genes associated with ALS and other neurodegenerative diseases. We show that there is time-of-day-dependent expression of ALS-linked RBP transcripts and rhythmic alternative splicing of genes involved in microtubule cytoskeleton organization, intracellular trafficking, and synaptic function. In addition, we demonstrate clock-dependent expression of ALS-linked RBP Ataxin-2 (ATXN2) in this neuronal subtype. Through in silico analysis of RNA sequencing (RNA-seq) data from sporadic ALS (sALS) patients, we find that gene expression profiles altered in disease correspond with rhythmic gene networks. Finally, we report that clock disruption through cholinergic neuron-specific deletion of clock activator BMAL1 (*i*) increases lumbar spinal cord motor neuron loss and sciatic nerve axon degeneration and (*ii*) drives time-of-day-dependent alternative splicing of genes associated with RNA processing, including genes encoding ALS-linked RBPs (e.g., *Matr3*, *Srsf7*, and *Ythdf2*). Thus, we propose that disruption of the cholinergic neuron circadian clock alters RNA metabolism and contributes to the neurodegenerative phenotypes observed in cells affected in ALS.

# Results

## Spinal cord cholinergic neuron rhythmic transcriptome is enriched for genes associated with ALS and other neurodegenerative diseases

Given that ALS disease fatality stems from cellular dysfunction occurring within acetylcholine-releasing LMNs, also known as spinal cord cholinergic neurons, we aimed to characterize the rhythmic transcriptome of this neuronal subtype. To elucidate the cholinergic neuron transcriptome, we first generated *ChAT-Cre*; *RiboTag$^{fx/+}$* mice, which possess a ribosomal hemagglutinin (HA) tag to enable the isolation and characterization of mRNAs specifically bound to cholinergic neuron ribosomes. We harvested and isolated RNA from WT spinal cord cholinergic neurons every 4 h across the 24-h day for RNA-seq (Figs 1A and S1A and B).

Rhythmic transcriptome profiling revealed 2,391 oscillating transcripts (~12% of genes) by Jonckheere-Terpstra-Kendall (JTK)_Cycle analysis, with most rhythmic transcripts reaching maximal abundance at Zeitgeber Time (ZT) 20 and 22 in the dark/active period, and a smaller cluster of genes reaching peak expression at ZT8 in the light/rest period (Fig 1B and C; Table S1A). Kyoto Encyclopedia of Genes and Genomes (KEGG) pathway analysis showed overall enrichment of genes associated with *(i)* neurodegenerative diseases (ALS, Parkinson's disease, Huntington's disease, prion disease, and spinocerebellar ataxia); *(ii)* protein synthesis and degradation (spliceosome, ribosome, protein processing in the endoplasmic reticulum, proteasome, and ubiquitin-mediated proteolysis); *(iii)* metabolism (oxidative phosphorylation, TCA cycle, and metabolic pathways); and *(iv)* synaptic transmission (GABAergic synapse, dopaminergic synapse, and long-term potentiation) (Fig 1D).

Notably, we found significant enrichment of the oscillating transcripts for KEGG pathway genes associated with ALS, including genes related to oxidative phosphorylation, autophagy, and the oxidative stress response (Fig 1E; Table S1B). In addition, 15 of the rhythmically expressed ALS genes were associated with the Gene Ontology (GO) term "cytoskeleton organization" (Fig 1F; Table S1B). In analyzing the genes with peak expression during the light and dark periods separately, we found 711 rhythmic genes with peak abundance during the light period (ZT0-12), and GO enrichment analysis revealed over-representation for RNA processing (e.g., mRNA splicing) and biosynthetic processes (e.g., translation) (Fig S2). In contrast, 1,680 rhythmic genes exhibited peak abundance during the dark period (ZT12-24), and GO analysis revealed enrichment for pathways associated with protein localization and transport (Fig S2). Our findings suggest that these processes are regulated by the circadian clock in cholinergic neurons.

## Circadian clock regulates expression of ALS-linked RBPs and alternative splicing in spinal cord cholinergic neurons

RBPs regulate alternative splicing by interacting with regulatory elements in pre-mRNA to guide spliceosome recruitment (Ule & Blencowe, 2019). Clues to the rhythmic expression of genes encoding ALS-linked RBPs in neuronal subtypes originated from earlier studies demonstrating oscillatory RBP abundance in various mouse CNS regions, including brain stem, suprachiasmatic nucleus (SCN), hypothalamus, and pituitary (Fig S3) (Pizarro et al, 2013). To determine whether RBP-mediated alternative splicing is regulated by the circadian clock in cholinergic neurons, we assessed rhythmic RBP expression and identified 70 oscillating transcripts that encode splicing-related RBPs, including five that overlap with rhythmically expressed ALS genes—*Fus*, *Hnrnpa2b1*, *Atxn2*, *Taf15*, and *Ncbp1* (Fig 2A; Table S1C). Based on these findings, we proposed that there is clock regulation of ALS-linked RBP expression and alternative splicing in this neuronal subtype. To test this hypothesis, we first analyzed RBP protein expression in primary rat spinal cord cholinergic neurons with siRNA knockdown of *Bmal1* and scramble (SCR) controls. Notably, we found a significant decrease in Ataxin-2 (ATXN2) with *Bmal1* loss (Fig 2B), which suggests that ATXN2 protein expression is clock-dependent. Given that we did not observe differential mRNA expression of *Atxn2* (or

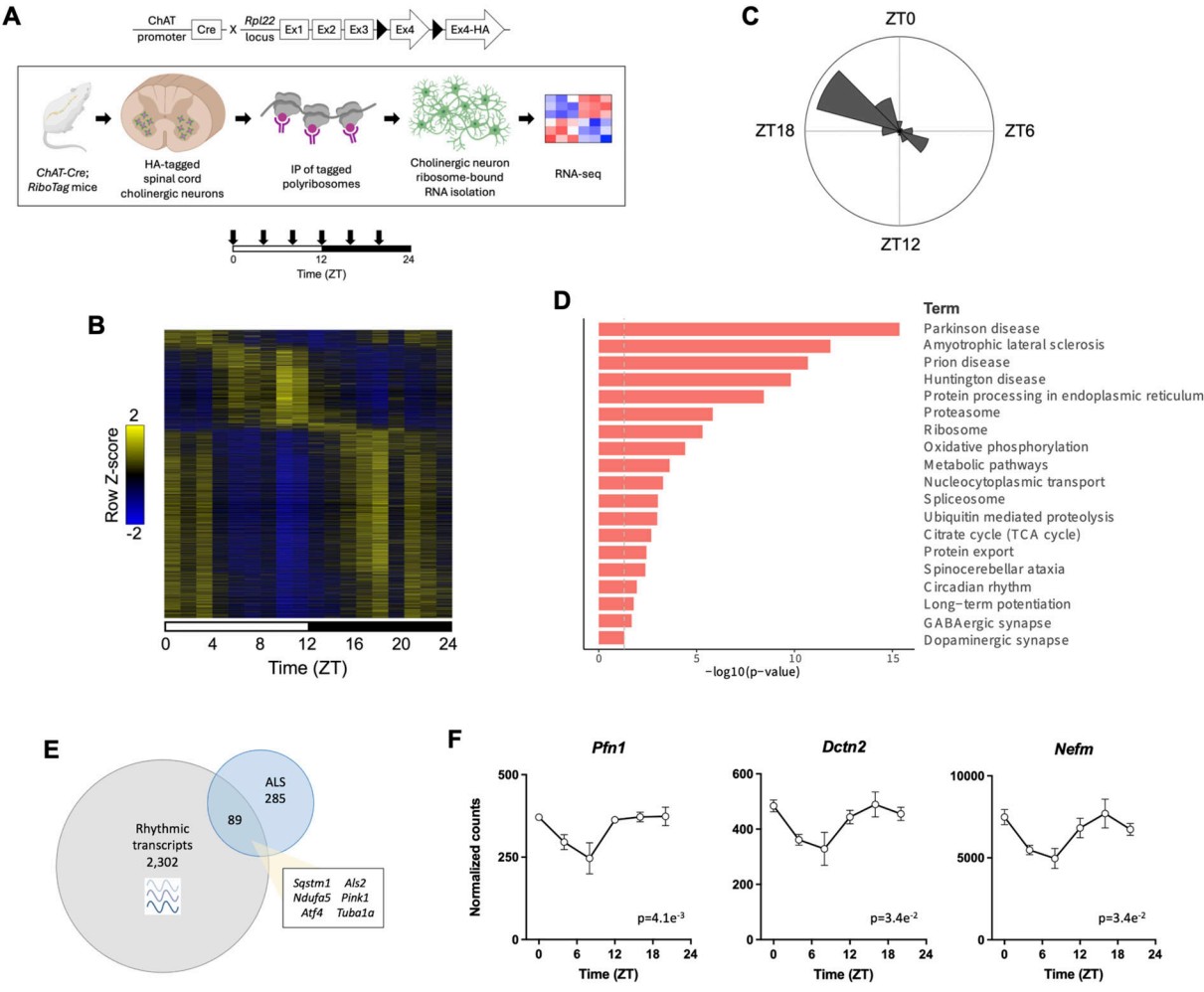

**Figure 1. Spinal cord cholinergic neuron rhythmic transcriptome is enriched for genes associated with ALS and other neurodegenerative diseases.**
**(A)** Schematic showing RNA isolation in WT spinal cord cholinergic neurons from 10-wk-old *ChAT-Cre;RiboTag*$^{fx/+}$ male mice collected every 4 h across the 24-h day for RNA-seq. Down arrows indicate sample harvesting times (ZT0, 4, 8, 12, 16, and 20). Created using BioRender. **(B)** Heatmap showing rhythmic cholinergic neuron mRNA expression (*n* = 3 per time point). **(C)** Radial histogram showing the number of oscillating genes that have peak expression within each 2-h window. **(D)** KEGG pathway analysis of all oscillating genes in cholinergic neurons. **(E)** Enrichment of KEGG pathway ALS genes among rhythmic transcripts in WT cholinergic neurons (adjusted *P* < 0.0001 in the hypergeometric test). **(F)** Examples of time-of-day-dependent expression of ALS-linked genes that regulate cytoskeleton dynamics, shown as normalized counts identified through DESeq2 (*n* = 3 per time point). Data are represented as the mean ± SEM. Rhythmicity was determined by JTK_Cycle analysis (adjusted *P*-value shown).

other rhythmically expressed ALS-linked RBP transcripts) with cholinergic neuron-specific *Bmal1* deletion, it is likely that this decrease in ATXN2 expression is a result of clock-driven post-translational events. In key studies performed in *Drosophila*, the ATXN2 homolog has been found to contribute to the regulation of circadian rhythms by activating the translation of core clock protein PERIOD (PER) (Lim & Allada, 2013; Zhang et al, 2013). Furthermore, a recent report demonstrated ATXN2 regulation of PER2 translation and oscillatory phase separation in mouse SCN (Zhuang et al, 2023).

We observed a bimodal distribution in peak phases of mRNA abundance of the oscillating splicing-associated RBPs 12 h apart at ZT8 and ZT20 (Fig 2C). To next determine whether alternative pre-mRNA splicing in cholinergic neurons is influenced by circadian time, we used replicate multivariate analysis of transcript splicing

(rMATS) to identify alternative splicing events in WT cholinergic neurons collected at 4-h intervals (Fig 1A), including skipped exons (SE), mutually exclusive exons (MXE), and alternative 3' and 5' splice sites (A3SS and A5SS). We calculated the percent-spliced-in (psi, Ψ) value for all common alternative splicing events in each of the three replicate samples at each time point and performed JTK_Cycle analysis to identify statistically significant oscillations in Ψ. This analysis revealed 11 rhythmic alternative splicing events, consisting of 8 SE, 1 MXE, 1 A3SS, and 1 A5SS (Fig 2D; Table S2A–D). Peak Ψ values also displayed a bimodal distribution 12 h apart (at ZT0-2 and ZT12-14), with most Ψ values peaking at ZT12 (Fig 2E). Sashimi plots, which illustrate exon inclusion levels by mapping reads across splice junctions, are shown throughout the circadian day to demonstrate rhythmic splicing of the representative gene *Aftph*, which encodes aftiphilin—a factor that enables clathrin

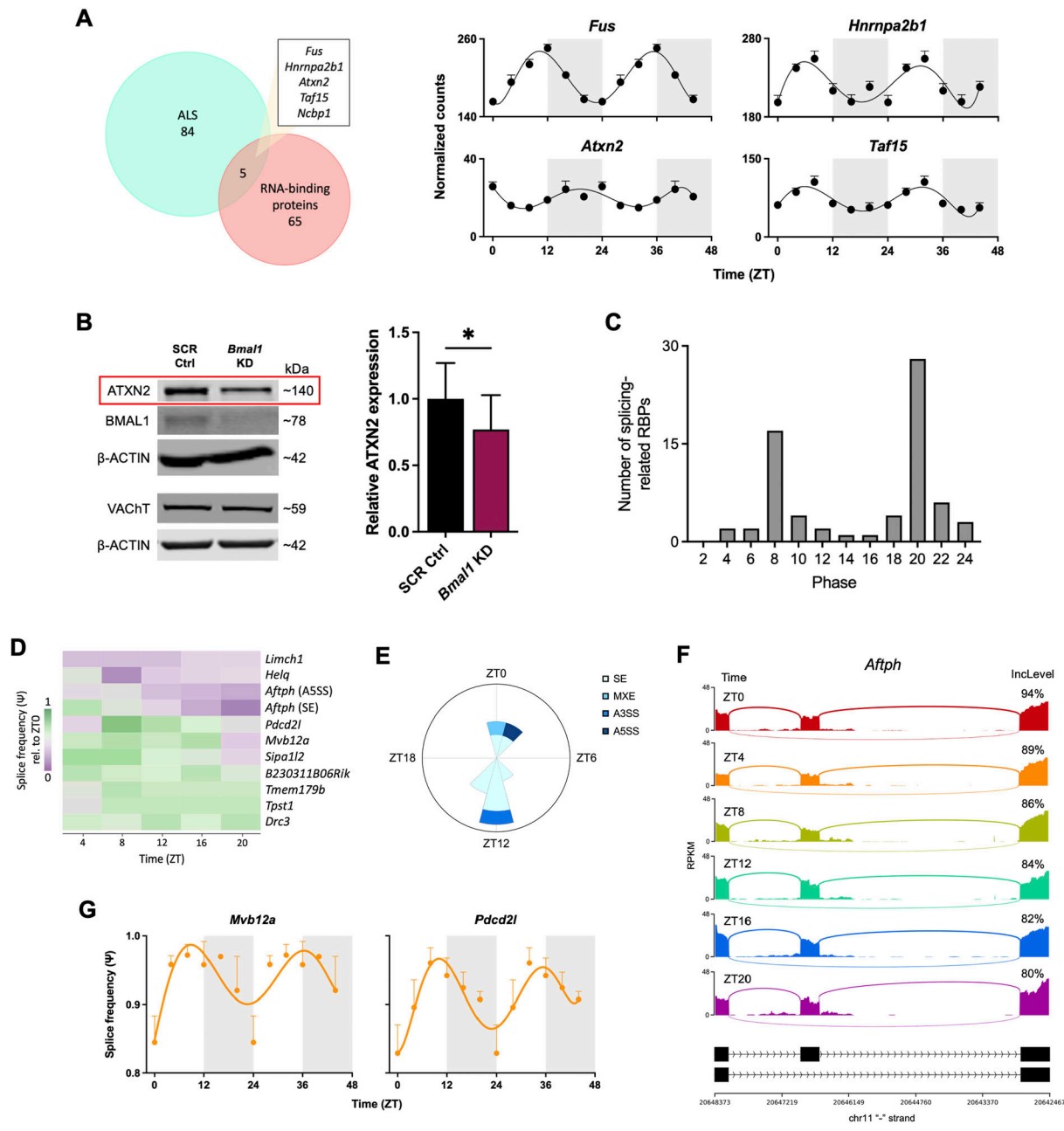

**Figure 2. Circadian clock regulates expression of ALS-linked RBPs and alternative splicing in spinal cord cholinergic neurons.**
**(A)** Rhythmically expressed genes that encode ALS-linked RNA-binding proteins (RBPs) in WT spinal cord cholinergic neurons from 10-wk-old *ChAT-Cre;RiboTag*^fx/+^ male mice, shown as normalized counts identified through DESeq2 (*n* = 3 per time point). **(B)** Representative immunoblot showing ATXN2 protein expression in primary rat spinal cord cholinergic neurons with siRNA knockdown of *Bmal1* versus scramble (SCR) control (*n* = 4). VAChT was used to confirm the integrity of cholinergic neurons. Statistical significance was calculated by unpaired two-tailed *t* test (**P* < 0.05). **(C)** Peak phase in the expression of oscillating RBPs in WT cholinergic neurons within the GO term "RNA splicing." **(D)** Heatmap showing rhythmically spliced genes in WT cholinergic neurons obtained using rMATS splicing analysis, represented as splice frequency (Ψ) relative to ZT0 (*n* = 3 per time point). **(E)** Radial histogram showing the number of each type of rhythmic splicing event in WT cholinergic neurons with peak Ψ occurring within each 2-h window. Radius corresponds to 4 splicing events. SE, skipped exon; MXE, mutually exclusive exon; A3SS and A5SS, alternative 3′ and 5′ splice sites. **(F)** Sashimi plots showing time-of-day-dependent alternative splicing of *Aftph* by SE every 4 h across the 24-h day in WT cholinergic neurons, with exon inclusion levels indicated at each time point. **(G)** Rhythmic splice frequency graphs of the genes that are both rhythmically expressed and rhythmically spliced in WT cholinergic neurons. For (A, G), data are represented as the mean ± SEM and double-plotted for better visualization. Rhythmicity was determined by JTK_Cycle analysis (adjusted *P*-value < 0.05). The shaded regions denote the dark/active time periods.

binding and intracellular transport as part of the AP-1 adaptor complex (Fig 2F) (Burman et al, 2005). We found that *Aftph* was rhythmically spliced by both SE and A5SS (Fig 2D; Table S2A and D).

LIMCH1 promotes actin stress fiber assembly and focal adhesion, and alternative splicing of *Limch1* is associated with muscle weakness and calcium-handling deficits (Lin et al, 2017; Penna et al, 2023). Our data revealed that both *Limch1* and dynein regulatory complex gene *Drc3* are rhythmically alternatively spliced by SE (Fig 2D; Table S2A). In addition, Rap GTPase-activating protein (RapGAP)-encoding gene *Sipa1l2* regulates synaptic function and vesicle trafficking at presynaptic terminals, and we found that it also displayed rhythmic altered splicing by SE (Fig 2D; Table S2A) (Andres-Alonso et al, 2019). Two genes exhibited both rhythmic mRNA expression and rhythmic splicing: vesicle trafficking gene *Mvb12a* and apoptosis gene *Pdcd2l* (Fig 2G) (Tsunematsu et al, 2010; Gao et al, 2022). We establish here that genes involved in microtubule cytoskeleton organization, intracellular trafficking, and synaptic function are rhythmically spliced in cholinergic neurons. Of note, a previous report demonstrated that rhythmically spliced genes in pancreatic islets were similarly associated with trafficking, exocytosis, and synaptic regulation (Marcheva et al, 2020). Taken together, these data demonstrate time-of-day-dependent expression of ALS-linked RBP transcripts as well as clock-dependent expression of ATXN2, and establish for the first time that the circadian clock regulates alternative splicing in cholinergic neurons.

### Cholinergic neuron circadian clock regulates sALS-associated cytoskeleton, ECM, and NMJ gene expression

The circadian clock regulates a number of physiological processes, including sleep/wake activity, hormone secretion, and glucose homeostasis (Lowrey & Takahashi, 2004; Rudic et al, 2004). Disrupting the clock through whole-body knockout or mutation of clock activators leads to metabolic dysfunction, accelerated aging, and neurodegenerative phenotypes (Garcia et al, 2000; Kondratov et al, 2006; Takahashi et al, 2008; Englund et al, 2009). Given the involvement of the cholinergic neuron clock in mediating neurodegenerative processes, we next sought to investigate the transcriptomic consequences of clock disruption through cholinergic neuron-specific deletion of clock activator BMAL1 at ZT8 (light period) and ZT20 (dark period), the times at which oscillating transcripts reached maximal abundance in opposing phases (Figs 1C, 3A, and S1A–C).

We found 48 upregulated genes and 21 downregulated genes in *ChAT-BKO* neurons (*Bmal1*-knockout cholinergic neurons isolated from *ChAT-Cre;RiboTag$^{fx/+}$;Bmal1$^{fx/fx}$* mice) compared with controls (WT cholinergic neurons isolated from *ChAT-Cre;RiboTag$^{fx/+}$* mice) at ZT8, and 74 upregulated genes and 12 downregulated genes in *ChAT-BKO* neurons at ZT20 (Fig 3B and D; Table S3A and B). KEGG, GO, and Reactome functional pathway analyses revealed over-representation for extracellular matrix (ECM) and cytoskeleton organization, as well as immune response terms among the upregulated genes at both time points (Fig 3C and E). Furthermore, deletion of *Bmal1* at ZT20 induced the expression of genes within the GO term "muscle system process" (Fig 3F), which suggests that clock disruption during the dark/active period may influence

neuromuscular junction (NMJ) dynamics. Enrichment for circadian rhythms appeared in the up- and down- regulated genes at both time points, as BMAL1 is a component of the core clock molecular feedback loop and its dysregulation influences the abundance of other clock genes (Table S3C). The cytoskeleton plays an important role in axonal transport and synaptic integrity, and cytoskeletal defects are prominent in ALS patients and disease models (Clark et al, 2016; Le Gall et al, 2020). Furthermore, ECM and cytoskeleton crosstalk is crucial for normal cell adhesion and migration processes, and both transcriptomic and proteomic studies have noted dysregulation of ECM biology in ALS samples (Geiger et al, 2001; Qiu et al, 2014; Lin et al, 2020; Milioto et al, 2024). The prevalence of cytoskeleton and ECM-associated genes within the differentially expressed genes in *Bmal1*-deficient cholinergic neurons suggests that clock disruption drives a restructuring that may contribute to similar deficits in axonal transport, cell adhesion, and synaptic integrity.

Notably, our data revealed an increase in gene expression of the ionotropic glutamate receptor *N*-methyl-D-aspartate (NMDA) subunit *Grin2c* in *ChAT-BKO* cholinergic neurons at ZT20 (Fig 3G). *Grin2c* was also found to be an RNA target of ALS-linked RBP FUS in mouse brain (Table S7) (Lagier-Tourenne et al, 2012). We previously showed that most rhythmic genes reach maximal abundance during the dark/active period and that there is enrichment of these genes for energy demanding processes (Figs 1C and S2); our findings here suggest that clock disruption at ZT20 may alter cholinergic neuron bioenergetic balance in a manner that increases glutamate demand. We further observed time-of-day-dependent expression of genes encoding NMDA receptor subunits *Grin1*, *Grina*, and *Grin2b* (Fig 3G), which may reflect how cholinergic neurons respond to fluctuations in glutamate release and demand throughout the day. Glutamate is the most abundant excitatory neurotransmitter in the CNS, and glutamate excitotoxicity—or the excess release of glutamate and overstimulation of glutamate receptors—is associated with motor neuron death in ALS and other neurodegenerative diseases (Van Den Bosch et al, 2006; Foran & Trotti, 2009). Thus, our findings may have important implications for the timing and dosage of ALS drugs, such as riluzole, which uses sodium channel blockade to inhibit glutamatergic neurotransmission and excitotoxicity (Nagoshi et al, 2015; Arnold et al, 2024).

To next determine whether transcriptomic findings in mouse cholinergic neurons can be translated to humans, we performed in silico analysis of RNA-seq data from ALS patients and found that gene expression profiles altered in disease correspond with rhythmic gene networks. In analyzing lumbar spinal cord motor neurons from 13 sALS patients and 8 healthy controls (Krach et al, 2018), we found 1,270 upregulated genes and 318 downregulated genes (Fig 4A; Table S4A). KEGG, GO, and Reactome functional pathway analyses revealed over-representation of genes associated with ECM, cytoskeleton organization, and immune response among the upregulated genes (Fig S4A), and enrichment of genes broadly linked to cytoskeleton dynamics and synaptic activation among the downregulated genes (Fig S4B).

Gene set enrichment analysis (GSEA) showed that the orthologous upregulated genes in *ChAT-BKO* mouse cholinergic neurons

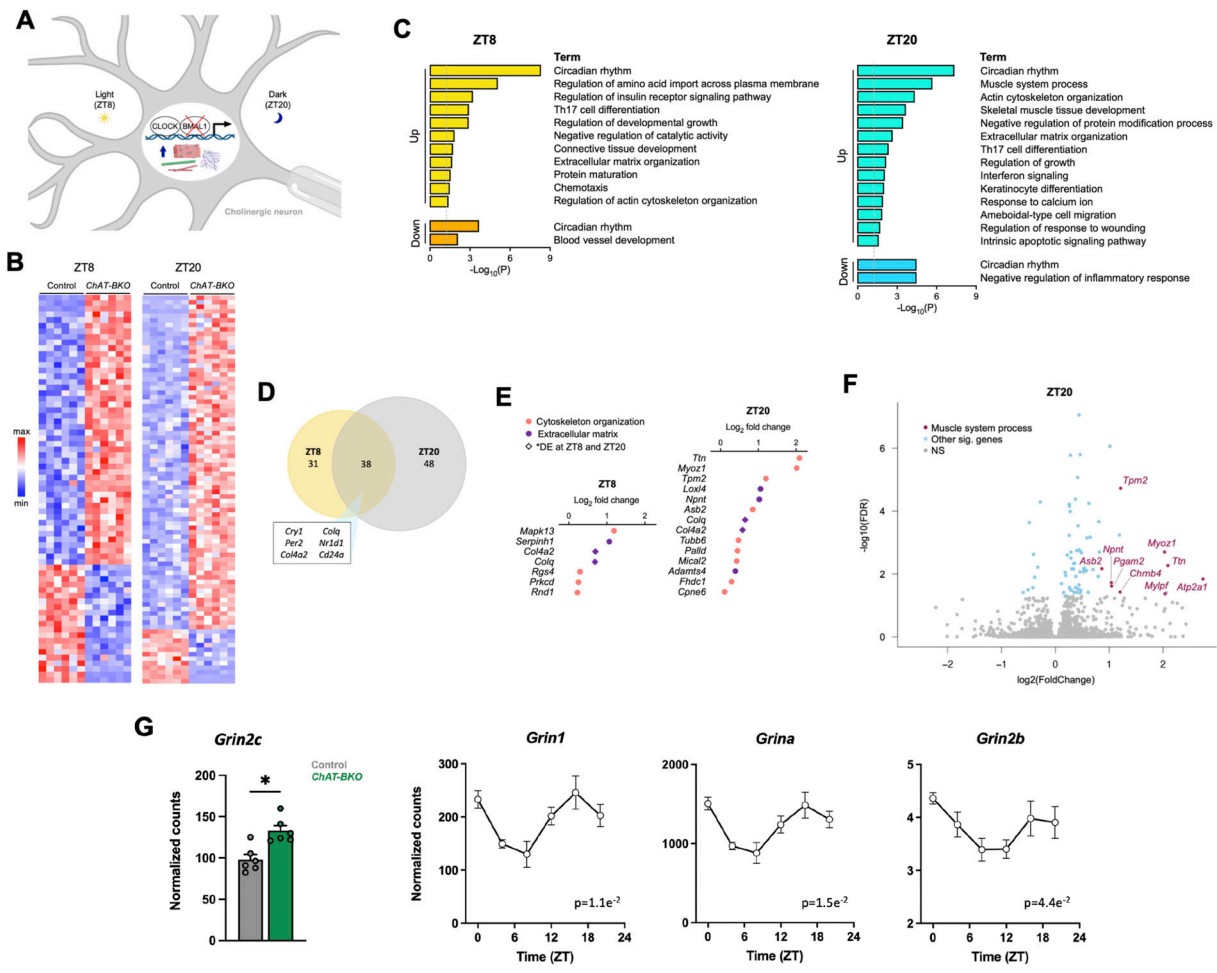

**Figure 3. Cholinergic neuron circadian clock regulates cytoskeleton, ECM, and NMJ gene expression.**
**(A)** Model of cholinergic neuron-specific deletion of BMAL1 during the light (ZT8) and dark (ZT20) periods to induce the expression of cytoskeleton, extracellular matrix (ECM), and neuromuscular junction (NMJ) genes. Created using BioRender. **(B)** Heatmaps showing differentially expressed genes between *ChAT-BKO* (*Bmal1*-knockout cholinergic neurons isolated from *ChAT-Cre;RiboTag*^fx/+^;*Bmal1*^fx/fx^ mice) and control (WT cholinergic neurons isolated from *ChAT-Cre;RiboTag*^fx/+^ mice) neurons from 6-mo-old male mice at ZT8 and ZT20 (*n* = 6 per condition). **(C)** KEGG, GO, and Reactome functional pathway analyses of genes that are up- or down-regulated in *ChAT-BKO* cholinergic neurons versus controls at ZT8 and ZT20. **(D)** Overlap of differentially expressed genes in *ChAT-BKO* cholinergic neurons at ZT8 and ZT20. **(E)** Increased expression of GO or literature-defined genes associated with cytoskeleton organization and ECM in *ChAT-BKO* cholinergic neurons at ZT8 and ZT20. **(F)** Volcano plot showing induction of NMJ genes within the GO term "muscle system process" in *ChAT-BKO* cholinergic neurons at ZT20. **(G)** Induction of glutamate receptor NMDA subunit *Grin2c* in *ChAT-BKO* cholinergic neurons at ZT20 (*n* = 6 per condition) and time-of-day-dependent expression of genes encoding other NMDA subunits (*n* = 3 per time point), shown as normalized counts identified through DESeq2. Data are represented as the mean ± SEM. Statistical significance was calculated using the Wald test with the Benjamini-Hochberg correction for multiple comparisons (*adjusted *P* < 0.05), and rhythmicity was determined by JTK_Cycle analysis (adjusted *P*-value shown).

at both ZT8 and ZT20 tended to be significantly upregulated in sALS patient LMNs as well (Fig 4B). Through hypergeometric testing, we found overlap and enrichment of several upregulated genes in *ChAT-BKO* mouse cholinergic neurons among upregulated genes in sALS patient LMNs, including ECM organization genes *COL4A2* and *COLQ*, actin cytoskeleton gene *PALLD*, core clock repressor gene *CRY1*, and fatty acid elongation factor *ELOVL7* (Fig 4C; Table S4B and C). Furthermore, our analysis revealed upregulation of actin filament stabilization gene *TPM2* and cytoskeleton-regulating gene *PLA2G7* with *Bmal1* loss at ZT20 (Lehtinen et al, 2017), as well as over-representation of these genes among the upregulated genes in sALS patient LMNs

(Fig 4D). We did not observe enrichment of downregulated genes in *ChAT-BKO* mouse cholinergic neurons among downregulated genes in sALS patient LMNs or enrichment of up- or down-regulated genes among sALS patient differentially expressed genes with opposite directionality (Table S4B). Our findings highlight the overlapping transcriptomic changes driven by clock disruption and neurodegenerative disease and reiterate the importance of ECM and cytoskeleton organization in neuronal health.

ALS-linked mutations disrupt RNA processing events, including mRNA axonal transport and local translation (Kapeli et al, 2017; Nagano & Araki, 2021). Local translation enables the

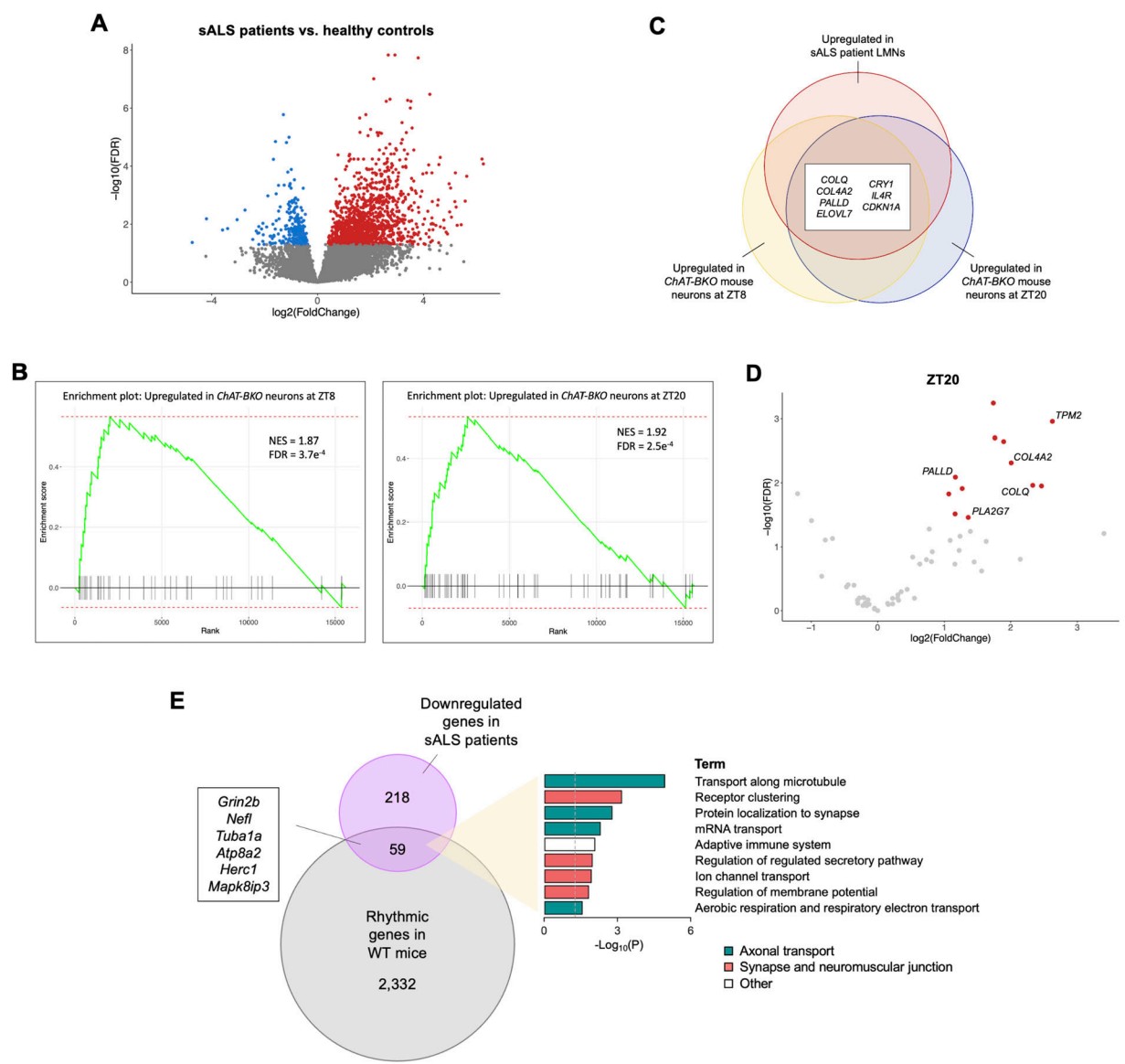

**Figure 4. Cholinergic neuron circadian clock regulates sALS-associated cytoskeleton, ECM, and NMJ gene expression.**
**(A)** Volcano plot showing differentially expressed genes between sporadic ALS (sALS) patient lumbar spinal cord motor neurons ($n$ = 13) and healthy controls ($n$ = 8). Re-analysis of data set from Krach et al (2018). **(B)** GSEA plots showing enrichment of orthologous upregulated genes in *ChAT-BKO* neurons (*Bmal1*-knockout cholinergic neurons isolated from 6-mo-old male *ChAT-Cre;RiboTag$^{fx/+}$;Bmal1$^{fx/fx}$* mice) at ZT8 and ZT20 ($n$ = 6 per condition) among rank-ordered genes in sALS patient lower motor neurons (LMNs) (normalized enrichment score [NES] and false discovery rate [FDR]–adjusted $P$-value shown). **(C)** Orthologous upregulated genes in *ChAT-BKO* mouse neurons at ZT8 and ZT20 that are over-represented among upregulated genes in sALS patient LMNs (adjusted $P$ < 0.01 in hypergeometric tests at both time points). **(D)** Volcano plot of orthologous upregulated genes in *ChAT-BKO* mouse neurons at ZT20 that are significantly enriched among upregulated genes in sALS patient LMNs. **(E)** Enrichment of orthologous sALS patient LMN downregulated genes (277 genes) associated with axonal transport and NMJ synaptic regulation among rhythmic genes in WT mouse cholinergic neurons (adjusted $P$ < 0.01 in hypergeometric test).

production of specific proteins when and where they are required, such as at synapses, where this process is critical for maintaining synaptic integrity and neurotransmission (Jung et al, 2020; Das et al, 2021). Notably, we found significant enrichment of 59 orthologous downregulated genes in sALS patient LMNs associated with axonal transport (e.g., *Nefl*, *Tuba1a*, and *Mapk8ip3*) and NMJ synaptic regulation (e.g., *Grin2b*, *Atp8a2*, and *Herc1*) among rhythmic genes in WT mouse cholinergic neurons (Fig 4E; Table S4D). Although our analysis

revealed 112 genes that were both upregulated in sALS patient LMNs and rhythmically expressed in WT mouse cholinergic neurons (Table S4E), there was no significant enrichment of orthologous sALS patient upregulated genes (1,183 genes) among the WT mouse rhythmic genes (adjusted $P$ = 0.99 in hypergeometric test). These data suggest that there is circadian regulation of sALS-associated processes on the mRNA level and support a role of the cholinergic neuron clock in mediating neurodegenerative phenotypes.

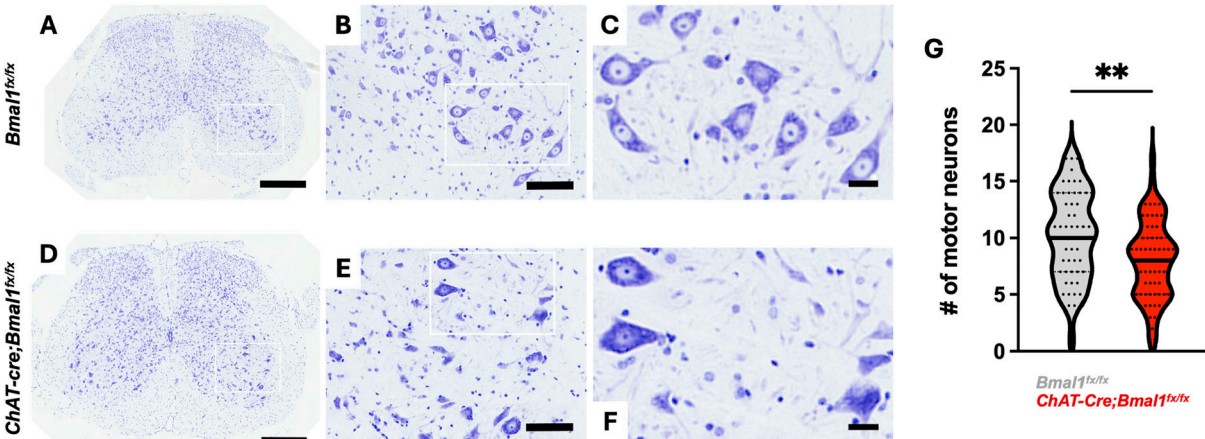

**Figure 5. Cholinergic neuron-specific *Bmal1* deletion drives lumbar spinal cord motor neuron loss.**
**(A, B, C, D, E, F)** Representative images of Nissl-stained lumbar spinal cord sections from 5- to 7-mo-old male mice with cholinergic neuron-specific *Bmal1* deletion (*ChAT-Cre;Bmal1^{fx/fx}* mice) (D, E, F) and controls (*Bmal1^{fx/fx}* mice) (A, B, C). Scale bars represent 400 $\mu$m (A, D), 100 $\mu$m (B, E), and 25 $\mu$m (C, F). **(G)** Quantification of motor neurons per lumbar spinal cord section in *ChAT-Cre;Bmal1^{fx/fx}* and *Bmal1^{fx/fx}* mice (*n* = 5–6). Data are represented as median with interquartile range. Statistical significance was calculated by unpaired two-tailed *t* test (**$P$ < 0.01).

## Cholinergic neuron-specific *Bmal1* deletion drives neurodegeneration in lumbar spinal cord and sciatic nerve but does not alter mouse metabolic indices

Extensive loss of LMNs in the spinal cord and brain stem is a key neuropathological feature of ALS (Grad et al, 2017). Based on our transcriptomic analysis indicating a role of the cholinergic neuron clock in mediating neurodegenerative processes, we hypothesized that clock disruption may affect motor neuron health and survival. To test this, we performed Nissl staining and morphometric quantification of LMNs, with the criteria for motor neuron designation being neurons located in the ventral horn with cell bodies between 30 and 70 $\mu$m in diameter, the presence of a multipolar structure, and a visible nucleolus. We found a significant decrease in the number of motor neurons in lumbar spinal cord sections from mice with cholinergic neuron-specific *Bmal1* deletion (*ChAT-Cre;Bmal1^{fx/fx}* mice) compared with those from controls (*Bmal1^{fx/fx}* mice) (20% decrease in median counts) (Fig 5A–G).

The sciatic nerve enables cholinergic neuron innervation of hindlimb skeletal muscle at the NMJ, and sciatic nerve axon damage is associated with muscle denervation and ALS disease progression (Gordon et al, 2009; Moloney et al, 2014). Given the involvement of the clock in synaptic transmission (Gizowski et al, 2016; Marcheva et al, 2020), we next sought to test whether clock disruption alters NMJ innervation through changes to the sciatic nerve. We performed toluidine blue staining and transmission electron microscopy (TEM) to visualize alterations in sciatic nerve morphology. We observed a significant increase in the number of severely degenerated axons in mice with *Bmal1* loss (122% increase in average counts), as characterized by expansion of the myelin sheath and axoplasmic abnormalities, such as shrinkage of the axoplasm (Fig 6A–D) (Essawy et al, 2011; Anderson et al, 2016; Muzio et al, 2020). Axon degeneration is an intrinsic self-destruction mechanism that leads to axon loss during injury or disease, and the NAD^+ hydrolase SARM1 was found to promote axonal

degeneration (Gerdts et al, 2015). Of note, our data revealed that *Sarm1* mRNA is rhythmically expressed in spinal cord cholinergic neurons (adjusted *P* < 0.001) (Table S1A), which suggests that there may be circadian regulation of this mediator of axon degeneration. In line with our transcriptomic analysis of *Bmal1*-deficient cholinergic neurons, we demonstrate morphological changes to sciatic nerve axons that may contribute to altered NMJ dynamics.

Circadian behavior includes rhythmic patterns of locomotor activity, respiration, and feeding (Bass & Takahashi, 2010), yet none of these parameters appeared to be altered with cholinergic neuron-specific *Bmal1* deletion (Fig S5A–C). Furthermore, we did not observe changes in mouse whole-animal body or spinal cord weights (Fig S5D and E). Therefore, we postulate that the cellular pathologies observed with cholinergic neuron clock disruption arise due to cell-autonomous defects.

## Cholinergic neuron-specific *Bmal1* deletion drives alternative splicing of ALS-linked RBPs and genes associated with RNA metabolism

Given that circadian timing plays a role in RBP expression and alternative splicing in cholinergic neurons, we next investigated whether clock disruption through *Bmal1* deletion during the light and dark periods drives changes in splicing. Our analysis of *ChAT-BKO* cholinergic neurons harvested at ZT8 and ZT20 revealed 60 alternative splicing events across 49 unique genes during the light period, consisting of 34 SE, 5 MXE, 12 A3SS, and 9 A5SS (Fig 7A and C; Table S5A–D), and 50 alternative splicing events across 41 unique genes during the dark period, consisting of 37 SE, 1 MXE, 4 A3SS, and 8 A5SS (Fig 7A and C; Table S6A–D). We uncovered differentially spliced genes with multiple transcript variants in cholinergic neurons with *Bmal1* loss, including ALS-associated genes *Stk16* (at ZT8) and *Atg10* (at ZT20), both of which underwent multiple exon skipping events (Fig 7B) (Lee et al, 2015; Zhao et al, 2017; Wang et al, 2019; García-García et al, 2021). Microtubule-

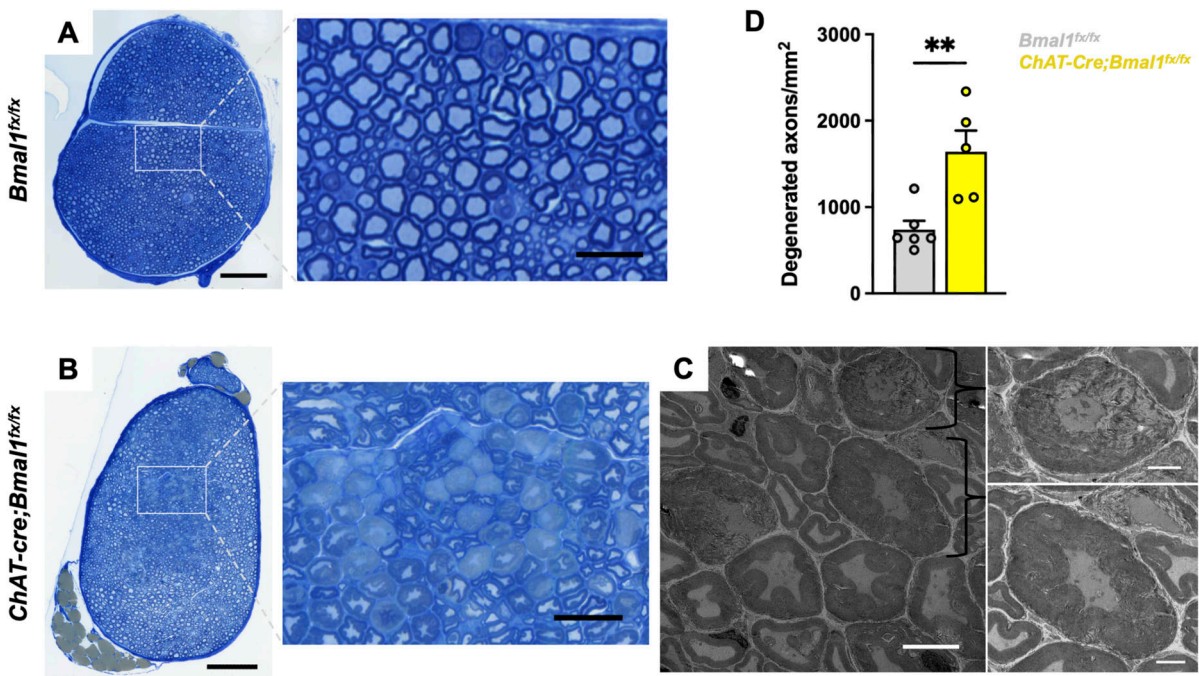

**Figure 6.  Cholinergic neuron–specific *Bmal1* deletion increases axon degeneration in sciatic nerve.**
**(A, B)** Representative images of toluidine blue-stained sciatic nerve sections from 5- to 7-mo-old male mice with cholinergic neuron-specific *Bmal1* deletion (*ChAT-Cre;Bmal1*^fx/fx^ mice) (B) and controls (*Bmal1*^fx/fx^ mice) (A). Scale bars represent 100 and 25 $\mu$m (magnified view). **(C)** Transmission electron microscopy (TEM) images of degenerated axons from *ChAT-Cre;Bmal1*^fx/fx^ mice shown in (B). Scale bars represent 5 and 2 $\mu$m (magnified view). **(D)** Quantification of degenerated axons per sciatic nerve section (measured in mm²) in *ChAT-Cre;Bmal1*^fx/fx^ and *Bmal1*^fx/fx^ mice (*n* = 5–6). Data are represented as the mean ± SEM. Statistical significance was calculated by unpaired two-tailed *t* test (***P* < 0.01).

associated protein gene *Ccdc74a* and histone variant-encoding gene *H2az2* were the only overlapping alternatively spliced genes between the two time points (Fig 7C).

Functional pathway analysis revealed enrichment for the regulation of mRNA stability and RNA splicing as the top terms among the genes that were differentially spliced in *ChAT-BKO* neurons at ZT8 and ZT20, respectively (Fig 7D), suggesting that *Bmal1* deletion in cholinergic neurons alters RNA metabolism. In addition to alternative splicing, RNA processing includes mRNA axonal transport along microtubules and local translation at synapses to enable efficient neurotransmission (Kapeli et al, 2017; Nagano & Araki, 2021). Deletion of *Bmal1* at ZT8 also resulted in altered splicing of several key genes related to mRNA stability, microtubule dynamics and transport, and postsynaptic organization (Fig 7E). Notably, *Ythdf2* was alternatively spliced by SE with *Bmal1* loss at ZT8 (Fig 7E). *Ythdf2* encodes the N6-methyladenosine (m6A) reader protein YTHDF2, which is an RBP that recognizes and directs m6A-modified RNAs; its dysregulation has been implicated in a number of neurological disorders (Song et al, 2024). Furthermore, YTHDF2 was found to accumulate in ALS spinal cord neurons and facilitate TDP-43–linked toxicity, and YTHDF2 knockdown prolonged the survival of human neurons with ALS-linked mutations (McMillan et al, 2023). In another example, we found that *Kif3a* was alternatively spliced by MXE with *Bmal1* loss at ZT8 (Fig 7E). *Kif3a* encodes the kinesin family molecular motor protein KIF3A, which was found to be reduced in sALS patient motor cortex (Pantelidou et al, 2007; Hirokawa et al, 2009). Interestingly, most of the

alternatively spliced genes were distinct between the two time points, and the alternatively spliced genes with *Bmal1* deletion at ZT20 were enriched for a different set of pathways, including ubiquitin-dependent protein catabolic process and response to cellular stress (Fig 7D). This demonstrates that there are time-of-day-dependent differences in cholinergic neuron circadian clock-mediated regulation of alternative splicing.

An early study comparing sALS lumbar spinal cord motor neurons with healthy controls noted little overlap between the differentially expressed and differentially spliced genes, positing that these cohorts of genes were largely distinct from one another (Rabin et al, 2010). Moreover, another report found minimal overlap between differentially expressed and differentially spliced genes in clock-disrupted pancreatic β-cells (0.6% and 4% overlap, respectively) (Marcheva et al, 2020). Consistent with these studies, we did not see any overlap between the differentially expressed and differentially spliced genes in *ChAT-BKO* cholinergic neurons at either time point. Collectively, our findings support that changes in splicing and mRNA abundance are regulated by distinct mechanisms that are downstream of the molecular clock. Of note, our gene expression and splicing analyses were conducted in mice in which we would expect to see the same neurodegenerative phenotypes observed in Figs 5 and 6, and thus, in addition to being a product of clock disruption, may also be affected by neurodegeneration.

Importantly, our data revealed altered splicing of genes that regulate mRNA stability and splicing in *ChAT-BKO* cholinergic

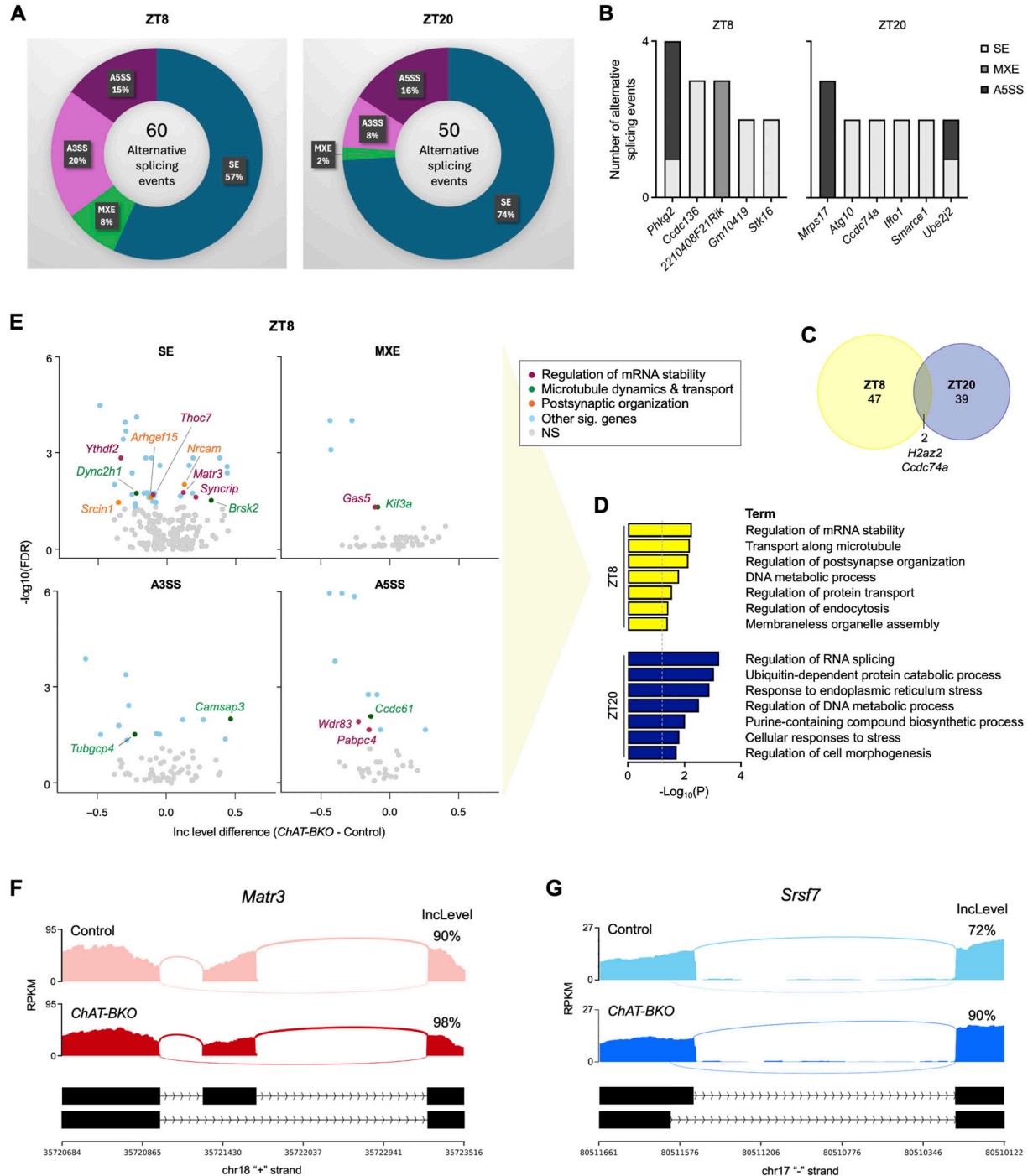

**Figure 7. Cholinergic neuron-specific *Bmal1* deletion drives alternative splicing of ALS-linked RBPs and genes associated with RNA metabolism.**
**(A)** Alternative splicing events in *ChAT-BKO* (*Bmal1*-knockout cholinergic neurons isolated from *ChAT-Cre;RiboTag*fx/+*;Bmal1*fx/fx mice) and control (WT cholinergic neurons isolated from *ChAT-Cre;RiboTag*fx/+ mice) neurons from 6-mo-old male mice at ZT8 and ZT20 determined by rMATS analysis (*n* = 6 per condition). SE, skipped exon; MXE, mutually exclusive exon; A3SS and A5SS, alternative 3′ and 5′ splice sites. **(B)** Differentially spliced genes with multiple transcript variants in *ChAT-BKO* cholinergic neurons at ZT8 and ZT20. **(C)** Overlap of alternatively spliced genes in *ChAT-BKO* cholinergic neurons at ZT8 and ZT20. **(D)** KEGG, GO, and Reactome functional pathway analyses of alternatively spliced genes in *ChAT-BKO* cholinergic neurons at ZT8 and ZT20. **(E)** Volcano plots of alternatively spliced genes by SE, MXE, A3SS, and A5SS in *ChAT-BKO* cholinergic neurons at ZT8, including several associated with mRNA stability, microtubule dynamics and transport, and postsynaptic organization. **(F, G)** Sashimi plots showing altered splicing of ALS-linked RBPs *Matr3* by SE in *ChAT-BKO* cholinergic neurons at ZT8 (F) and *Srsf7* by A5SS in *ChAT-BKO* cholinergic neurons at ZT20 (G), with exon inclusion levels indicated.

neurons, including RBP-encoding genes *Matr3*, *Syncrip*, and *Thoc7* at ZT8 (Fig 7E) and *Srsf7*, *Snrnp70*, and *U2af2* at ZT20. Matrin-3 (MATR3) is an ALS-linked DNA- and RNA-binding protein that plays key roles in chromatin organization, RNA processing, and mediating NMDA-induced neuronal cell death (Giordano et al, 2005; Salton et al, 2011; Coelho et al, 2015). It functions as a repressor of exon inclusion during alternative splicing, and mutations have been associated with increased cryptic exon inclusion and inhibition of mRNA nuclear export (Johnson et al, 2014b; Coelho et al, 2015; Boehringer et al, 2017; Uemura et al, 2017; Khan et al, 2024). We found that *Matr3* was alternatively spliced in cholinergic neurons with *Bmal1* deletion during the light period with increased inclusion of a skipped exon (Fig 7F; Table S5A) corresponding to exon 16 in the human genome (GRCh38.p14; ENST00000509990.5; chr5:139273752-139329404) (Dyer et al, 2025). Interestingly, a recent case study linked duplications in *Matr3* exons 15 and 16 to ALS (Caputo et al, 2022). Our analysis further revealed that ALS-linked RBP serine/arginine-rich splicing factor 7 (SRSF7) was alternatively spliced by A5SS in cholinergic neurons with *Bmal1* deletion during the dark period (Fig 7G; Table S6D), with changes corresponding to an increase in the inclusion level of an exon 5 coding variant in the human genome (GRCh38.p14; ENST00000409276.5; chr2:38744893-38751358) (Dyer et al, 2025). Of note, SRSF7 depletion was found to decrease the abundance of stathmin-2 (STMN2), an ALS-associated microtubule binding protein important for axon maintenance and regeneration after injury (Klim et al, 2019; López-Erauskin et al, 2024; Beccari et al, 2025; Wang et al, 2025).

We establish that there is circadian clock regulation of the splicing of RBPs themselves—including RBPs associated with ALS—and that because the alternatively spliced RBPs at ZT8 and ZT20 are distinct, this process is time-of-day-dependent. Importantly, we demonstrate that ALS-linked RBPs are rhythmically expressed in WT cholinergic neurons and that, with *Bmal1* deletion, ALS-linked RBPs are differentially spliced. Given that RBP mutations are a hallmark of ALS and other neurological disorders, it would be interesting to consider how clock-dependent gene expression and splice variants of these RBPs contribute to their own dysregulation and their control of alternative splicing in health and disease.

## Discussion

Alternative splicing plays a crucial role in the generation of transcriptomic and proteomic diversity, especially in neuronal cells, whose complex and specialized functions render them particularly vulnerable to disruption. The circadian clock adds another layer of complexity to this process by modulating post-transcriptional events based on the time-of-day-specific energetic demands of each cell. Given the length of motor neuron axons and the extensive distances RNA cargo has to travel, motor neurons are uniquely susceptible to disturbances in RNA processing, including disruptions to axonal transport and local synaptic translation. Disrupted homeostasis of RBP mediators of RNA metabolism through mutation often results in neurological and neurodegenerative disorders, such as ALS. In healthy cells, RBP-bound mRNA is packaged into ribonucleoprotein (RNP) granules that

are transported by motor proteins (e.g., dyneins and kinesins) along microtubules to axon terminals for local translation (Maday et al, 2014). Local translation of targeted mRNAs enables the production of specific proteins when and where they are needed, such as at neuronal synapses to maintain synaptic function (Jung et al, 2020; Das et al, 2021). Translation is also mediated by cellular stress, and RBP mutations have been shown to promote the formation of aberrant RNP stress granules—membraneless organelles that undergo phase transitions from transient liquid droplets to irreversible solid states (Kiebler & Bassell, 2006; Lin et al, 2015). These phase transitions often prevent RBPs from carrying out their normal functions.

Dysregulation of ALS-linked RBPs has been shown to disrupt multiple aspects of RNA metabolism. For example, ALS-linked FUS mutations were found to alter FUS regulation of a motor neuron survival factor SMN and inhibit its transport to axons (Groen et al, 2013), as well as impair RNP granule function by inducing phase transitions (Murakami et al, 2015; Patel et al, 2015). Furthermore, ALS-linked mutations in TAF15 were found to increase the number of TAF15-positive axonal and dendritic stress granules (Couthouis et al, 2011), and inhibition of TAF15 methylation resulted in the accumulation of TAF15 in TIA1-positive stress granules (Jobert et al, 2009). Our investigation into the link between two central evolutionarily conserved processes, the circadian clock transcriptional program and alternative splicing, yields important insight into time-of-day-dependent mechanisms of RBP regulation and RNA metabolism in cholinergic neurons. We demonstrate clock-dependent expression of the RBP ATXN2, a known regulator of the circadian clock (Lim & Allada, 2013; Zhang et al, 2013). In connection to neurodegenerative disease, polyglutamine (polyQ) repeat expansions in *ATXN2* cause spinocerebellar ataxia type 2 (SCA2) and increase the risk for development of ALS (Elden et al, 2010; Nussbacher et al, 2019; Glass et al, 2022). Pathogenic variants have been shown to aberrantly sequester TDP-43, inhibit mRNA transport and local translation, and reduce motor function and NMJ area (Vieira de Sá et al, 2024; Wijegunawardana et al, 2025). Of note, reduction of ATXN2 in transgenic mice was found to decrease TDP-43 aggregation and improve motor function and survival (Becker et al, 2017).

While our data support that neurodegenerative processes are mediated by the circadian clock in spinal cord cholinergic neurons—the cell-type of ALS disease pathogenesis—further investigation into how *Bmal1* loss influences the localization and function of ALS-linked RBPs (e.g., TDP-43 and FUS) would help establish a link between clock disruption and ALS-specific pathophysiology. In addition, functional analyses would be needed to inform causal relationships. Our work and previous studies demonstrating circadian regulation of alternative splicing in neurons and clock-dependent neurotransmission support that there may be clock-mediated RBP regulation of neurotransmitter release at motor neuron NMJ synapses (Shinohara et al, 2002; Gizowski et al, 2016; Wang et al, 2018; Marcheva et al, 2020). Cholinergic neurons release the neurotransmitter acetylcholine as a chemical messenger to transmit signals to other neurons and muscle cells in order to coordinate processes such as muscle contraction; acetylcholinesterase (AChE) is an enzyme that plays key roles in NMJ formation and acetylcholine breakdown during

signal termination (Fernandez et al, 1986; Campanari et al, 2016). Decreased acetylcholine release because of motor neuron degeneration, as well as decreased AChE activity in muscle and cerebrospinal fluid, has previously been reported in ALS patients (Fernandez et al, 1986; Campanari et al, 2016; Drannik et al, 2017). Given the neurodegenerative phenotypes we observed along with our transcriptomic data demonstrating altered NMJ dynamics with *Bmal1* deletion, it would be interesting to explore functional changes in acetylcholine and AChE levels in clock-disrupted spinal cord cholinergic neurons, especially in relation to changes in the rhythmically expressed ALS-linked RBPs (i.e., *Fus*, *Hnrnpa2b1*, *Atxn2*, *Taf15*, and *Ncbp1*).

We show that 70 splicing-related RBP transcripts are rhythmically expressed in spinal cord cholinergic neurons, including several that are directly linked to ALS (see above). Future studies should address whether any of these rhythmic RBPs regulate the alternatively spliced transcripts observed with *Bmal1* deletion in this neuronal subtype. Previous cross-linking and immunoprecipitation followed by sequencing (CLIP-seq) studies and variations have identified RNA targets of disease-linked RBPs in neuronal tissue and cells, many of which we found to be rhythmically expressed or spliced in WT cholinergic neurons or differentially regulated in cholinergic neurons with *Bmal1* deletion (Table S7). For instance, we note that actin stress fiber gene *Limch1* is rhythmically spliced in both cholinergic neurons and an overlapping target of rhythmically expressed ALS-linked RBPs FUS, TAF15, and hnRNPA2B1 (Table S7) (Ibrahim et al, 2013; Nakaya et al, 2013; Martinez et al, 2016), suggesting that one or more of these RBPs may directly regulate its RNA processing and protein function in a time-of-day-dependent manner. In another example, *Snrnp70* encodes an essential small nuclear ribonucleoprotein (snRNP) component of the spliceosome and is a target of both TAF15 and hnRNPA2B1 (Wahl et al, 2009; Ibrahim et al, 2013; Martinez et al, 2016; Nikolaou et al, 2022). Our results demonstrate that *Snrnp70* is rhythmically expressed in WT cholinergic neurons and differentially spliced in cholinergic neurons with *Bmal1* deletion (Table S7), raising the possibility that circadian clock disruption alters TAF15 or hnRNPA2B1 regulation of *Snrnp70* in cholinergic neurons. Further insight into cell-type-specific RBP-RNA interactions would inform how clock-dependent mechanisms contribute to alterations in RNA metabolism observed with neurological disease and provide insight into novel therapeutic avenues.

# Materials and Methods

### Animals

All animal procedures were conducted in accordance with regulations of the Institutional Animal Care and Use Committee at Northwestern University. All experiments were conducted using C57BL/6J mice maintained on a 12:12-h light/dark cycle. WT cholinergic neuron isolation was achieved by crossing *ChAT-Cre* mice (gift from L. Parisiadou, Northwestern University; JAX #006410) (Rossi et al, 2011) with *RiboTag$^{fx/fx}$* mice (obtained from Jackson Laboratories #029977) (Sanz et al, 2009) to generate *ChAT-Cre;*

*RiboTag$^{fx/+}$* mice. These experiments were performed using 10-wk-old male mice. Cholinergic neuron-specific deletion of *Bmal1* was achieved by crossing *Bmal1$^{fx/fx}$* mice (gift from J. Bass, Northwestern University) (Johnson et al, 2014) with *ChAT-Cre* or *RiboTag$^{fx/fx}$* mice to ultimately generate *ChAT-Cre;Bmal1$^{fx/fx}$* mice (along with *Bmal1$^{fx/fx}$* littermate controls) and *ChAT-Cre; RiboTag$^{fx/+}$;Bmal1$^{fx/fx}$* mice (along with *ChAT-Cre;RiboTag$^{fx/+}$* controls). These experiments were performed using 5- to 7-mo-old male mice to assess the added insult of older age on phenotype and transcriptional program.

### Activity, feeding, and weight measurements

Mice were individually housed in Promethion Core metabolic cages (Sable Systems International) where they were allowed free access to food and water and acclimated for 4 d before data collection. Locomotor activity (measured by beam breaks), respiratory exchange ratio (RER) (measured by the rate of carbon dioxide emission [$VCO_2$] divided by the rate of oxygen consumption [$VO_2$]), food intake, and body weight were continuously monitored for 3 d. To obtain spinal cord measurements, fresh spinal cord tissue was isolated, weighed, and normalized to mouse whole-body weight.

### Protein gel electrophoresis and immunoblotting

Whole-cell lysates were prepared in RIPA lysis and extraction buffer (89900; Thermo Fisher Scientific) supplemented with 1X Protease Inhibitor Cocktail (P8340; Sigma-Aldrich). Protein levels were quantified using the Pierce BCA protein assay kit (A65453; Thermo Fisher Scientific). Protein extracts were then subjected to SDS-PAGE gel electrophoresis and transferred to nitrocellulose membranes (1620115; Bio-Rad). Primary antibodies used were BMAL1 (1:1,000 dilution; D2L7G; Cell Signaling), ATXN2 (1:2,000 dilution; 21776-1-AP; Proteintech), VAChT (1:1,000 dilution; MA5-27662; Invitrogen), and *β*-ACTIN (1:5,000 dilution; NB600501; Novus Biologicals). Species-specific IRDye secondary antibodies were used (1:10,000 dilution; LI-COR Biotech LLC).

### Primary spinal cord motor neuron culture and siRNA transfection

Embryonic day 15 (E15) rat spinal cord was isolated and dissociated by incubation with 0.25% trypsin–EDTA (25200056; Gibco) for 15 min at 37°C. Samples were then centrifuged at 300 RCF for 10 min in Neurobasal media (21103049; Gibco) containing B-27 (A3582801; Gibco), 2.5% DNase I, and 4% BSA. Cells were resuspended by gentle up and down pipetting. The spin-resuspension process was then repeated. Motor neurons were enriched by adding 1 ml BSA to the bottom of the resuspension tube. Cultures were then plated onto poly-lysine-coated plates and maintained in conditioned Neurobasal media containing 10% horse serum. On day in vitro (DIV) 1, primary neurons were transfected with either three siRNAs targeting the knockdown of *Bmal1* in equal proportion (s131939-41; Thermo Fisher Scientific) or Silencer Select Negative Control siRNA (4390843; Invitrogen), using Lipofectamine RNAiMAX transfection reagent (13778150; Invitrogen) and reduced serum Opti-MEM I (31985062; Gibco) in antibiotic-free media. On DIV4, cultures were treated with 5 *μ*M AraC (C6645; Sigma-Aldrich) for 48 h to arrest

astrocyte proliferation (Mojsilovic-Petrovic et al, 2009). On DIV14, neurons were collected for immunoblotting (as described above).

## Lumbar spinal cord immunohistochemistry

Mice were anesthetized with intraperitoneal injection of Euthasol (390 mg/ml pentobarbital sodium, 50 mg/ml phenytoin sodium), and then transcardially perfused with PBS followed by 4% PFA in 0.4 M phosphate buffer (pH 7.4). Lumbar spinal cord was isolated, postfixed in the same fixative overnight at 4°C, and paraffin-embedded. 6-$\mu$m-thick sections were cut and mounted onto slides. Samples were then deparaffinized and hydrated before antigen retrieval by treatment with Antigen Decloaker (CB910M; Biocare Medical) in a high-pressure chamber for 20 min at 121°C (Deng et al, 2011). For immunohistochemistry, endogenous peroxidase activity was inhibited using 2% hydrogen peroxide. Samples were blocked for nonspecific binding by incubation with 1% BSA for 1 h at RT, and subsequently incubated with BMAL1 primary antibody (1:500 dilution; NB100-2288; Novus Biologicals) in a humidified chamber overnight at 4°C. The next day, samples were incubated with species-specific Alexa Fluor secondary antibody (1:500 dilution; Invitrogen) in the dark for 1 h at RT. Finally, samples were mounted with ProLong Gold Antifade Mountant (P36930; Invitrogen) and imaged on a DMI4000B confocal microscope (Leica Microsystems).

## Lumbar spinal cord Nissl staining and morphometric quantification

Mice were anesthetized with intraperitoneal injection of Euthasol (390 mg/ml pentobarbital sodium, 50 mg/ml phenytoin sodium), and then transcardially perfused with PBS followed by 4% PFA in 0.4 M phosphate buffer (pH 7.4). Lumbar spinal cord was isolated, postfixed in the same fixative overnight at 4°C, and paraffin-embedded. 6-$\mu$m-thick sections were cut and mounted onto slides. Samples were then deparaffinized and hydrated, before staining with heated Cresyl Violet acetate (C5042; Sigma-Aldrich) solution for 10 min and differentiation in 90% ethanol for 20 min. Finally, samples were dehydrated in ethanol, cleared in xylene, and mounted with DPX Mountant (06522; Sigma-Aldrich). Tile-stitched images at 20× magnification were acquired using TissueFAXS PLUS (TissueGnostics) to obtain whole spinal cord cross sections. After images were blinded, motor neurons were counted in every sixth section in an average of 11 sections per animal. Criteria for motor neuron designation were neurons located in the ventral horn with cell bodies between 30 and 70 $\mu$m in diameter, the presence of a multipolar structure, and a visible nucleolus.

## Sciatic nerve transmission electron microscopy and morphometric quantification

Mice were anesthetized with intraperitoneal injection of Euthasol (390 mg/ml pentobarbital sodium, 50 mg/ml phenytoin sodium), and then transcardially perfused with PBS followed by 4% PFA in 0.4 M phosphate buffer (pH 7.4). The sciatic nerve was isolated and immersion-fixed in 0.1 M sodium cacodylate buffer (pH 7.35) containing 2.5% glutaraldehyde and 2% PFA overnight at 4°C. Tissue was post-fixed in 2% osmium tetroxide, en bloc stained with

3% uranyl acetate, and dehydrated in ascending grades of ethanol followed by propylene oxide. Samples were then embedded in EMbed 812 Resin (14900; Electron Microscopy Sciences) and cured overnight in a 60°C oven. 1-$\mu$m-thick sections were cut using the EM UC6 Ultramicrotome (Leica Microsystems) and stained with toluidine blue O. Tile-stitched images at 40× magnification were acquired using TissueFAXS PLUS (TissueGnostics) to obtain whole nerve cross sections. After images were blinded, severely degenerated axons, characterized by expansion of the myelin sheath and axoplasmic abnormalities (e.g., shrinkage of the axoplasm) (Essawy et al, 2011; Anderson et al, 2016; Muzio et al, 2020), were counted within each nerve section and divided by the total area of the section (in mm$^2$). For transmission electron microscopy (TEM), ultrathin 70-nm sections were mounted onto mesh copper grids and stained with uranyl acetate and Reynolds lead citrate. Images at 1,400× and 2,900× magnification levels were captured at 80 kV using Tecnai G2 Spirit (FEI Company) and Tecnai Imaging and Analysis Software.

## Cholinergic neuron ribosome immunoprecipitation and RNA isolation

Spinal cord tissue was isolated and snap-frozen from 10-wk-old _ChAT-Cre;RiboTag$^{fx/+}$_ mice at ZT0, 4, 8, 12, 16, and 20 or from 6-mo-old _ChAT-BKO_ (_ChAT-Cre;RiboTag$^{fx/+}$;Bmal1$^{fx/fx}$_) mice and controls (_ChAT-Cre;RiboTag$^{fx/+}$_) at ZT8 and ZT20. All samples for each experiment were processed on the same day. Homogenization buffer consisting of 1% sodium deoxycholate, 50 mM Tris (pH 7.5), 100 mM KCl, 12 mM MgCl$_2$, 1% Tergitol (NP-40), 1 mM dithiothreitol (DTT), 200 U/ml RNasin Ribonuclease Inhibitor (N2115; Promega), 1 mg/ml heparin, 100 $\mu$g/ml cycloheximide, and Protease Inhibitor Cocktail (P8340; Sigma-Aldrich) was made fresh and prechilled. One spinal cord was placed in a 2-ml Dounce homogenizer vessel on ice with 2 ml homogenization buffer, and homogenized with 30 strokes of "pestle A" followed by 30 strokes of "pestle B." The homogenate was then centrifuged at 10,000 RCF for 10 min at 4°C. TRIzol reagent (15596026; Invitrogen) was added to 80 $\mu$l of the spun homogenate and frozen at −80°C as pooled Input (RNA from whole spinal cord). 800 $\mu$l of the remaining non-pelleted homogenate was incubated with 4 $\mu$l of anti-HA antibody (901514; BioLegend) for 4 h at 4°C with end-over-end rotation. Samples were then incubated with 200 $\mu$l of pre-washed protein A/G magnetic beads (Pierce, 88803; Thermo Fisher Scientific) overnight at 4°C with end-over-end rotation. High salt buffer consisting of 50 mM Tris (pH 7.5), 300 mM KCl, 12 mM MgCl$_2$, 1% NP-40, 0.5 mM DTT, and 100 $\mu$g/ml cycloheximide was made fresh and prechilled. Samples were placed on a magnetic stand on ice and washed three times with high salt buffer (800 $\mu$l per wash) to remove non-specific binding from the immunoprecipitate (IP) product. After careful removal of high salt buffer, samples were treated with 700 $\mu$l of TRIzol reagent and frozen at −80°C until RNA was extracted using the Direct-zol RNA MicroPrep kit (R2062; Zymo Research Corp.) following the manufacturer's instructions.

## Quantitative real-time PCR

After RNA isolation, cDNA was synthesized using the High-Capacity cDNA Reverse Transcription Kit (4368813; Applied Biosystems).

Quantitative real-time PCR (qPCR) was performed with iTaq Universal SYBR Green Supermix (1725124; Bio-Rad) using CFX384 Touch Real-Time PCR System (Bio-Rad) and analyzed using Bio-Rad CFX Manager Software (v3.1). Relative expression levels normalized to $β$-actin (steady between conditions) were determined using the comparative CT method. All primer sequences are listed in Table S8.

### Library preparation and RNA sequencing

After RNA isolation, RNA quality was assessed by Bioanalyzer (Agilent) and those with an RNA Integrity Number (RIN) greater than 7 were used for library prep. Libraries were constructed from 100 or 300 ng of RNA (for rhythmic or differential analyses, respectively) using the Illumina Stranded mRNA Prep kit (20040534; Illumina) according to the manufacturer's instructions. Libraries were quantified by both Bioanalyzer (Agilent) and qPCR-based quantification (Kapa Biosystems). 150-bp paired-end sequencing was performed on a NovaSeq X Plus (Illumina) to a minimum depth of 30 M reads. Sequences were aligned to the mm10 transcriptome with STAR (v2.7.5) using the –quantMode Transcriptome and GeneCounts options. Gene counts were obtained from mm10 annotations (Ensembl 98) using rsem-calculate-expression (v1.3.0).

### Differential expression, differential splicing, and rhythmicity analyses

Differential mRNA expression analysis was performed using DESeq2 (v1.32.0) after removal of genes with < 10 assigned counts. Differentially expressed transcripts were defined as having a false discovery rate (FDR)–adjusted $P$-value < 0.05. Alternative splicing analysis was performed using replicate multivariate analysis of transcript splicing (rMATS) (v4.0.2) (Shen et al, 2014) to quantify reads across annotated splice junctions present in the mm10 assembly. Differentially spliced transcripts were defined as having an FDR-adjusted $P$-value < 0.05 and a minimum exon inclusion level difference > 10%. Rhythmic splicing analysis was performed by first calculating psi values for all common alternative splicing events at each time point relative to ZT0. Rhythmic mRNA expression and rhythmic splice frequency throughout the day were assessed by JTK_Cycle analysis (Hughes et al, 2010), allowing for period lengths of 20, 24, or 28 h to account for 4-h sampling frequency. Rhythmicity was determined by an FDR-adjusted $P$-value < 0.05. KEGG, GO, and Reactome pathway enrichment analyses were performed using limma and biomaRt (v2.48.2), ShinyGO (v0.77) (Ge et al, 2020), Metascape (Zhou et al, 2019), and Cytoscape (Shannon et al, 2003). Certain plots were made in R (v4.0.3) using ggplot2 (v3.3.0), gplots (v3.1.1), pheatmap (v1.0.12), and fgsea (v1.16.0).

### Statistical analysis

Results are presented as the mean ± SEM unless otherwise indicated. Statistical analyses were performed by unpaired two-tailed $t$ test or analysis of variance (ANOVA) with corrections for multiple comparisons, where applicable. Hypergeometric tests were used to determine enrichment of genes from one dataset within another. The Wald test with the Benjamini-Hochberg correction for multiple comparisons was used for statistical analysis of normalized count data from DESeq2. $P$-values < 0.05 were considered statistically significant. Detailed information on genotype, sample size, and $P$-value can be found within individual figures and figure legends.

## Data Availability

All raw and processed sequencing data generated in this study have been deposited in the NCBI Gene Expression Omnibus under the accession number GSE297457.

## Supplementary Information

## Acknowledgements

We thank C Peek, M Perelis, H-X Deng, and members of the Kalb and Bass laboratories for helpful discussion. We also thank B Weidemann for bioinformatics training, Y Shi and H-X Deng for spinal cord experimental techniques, and L Reynolds Jr. for TEM assistance. We thank the Northwestern University Center for Genetic Medicine NUSeq Core, the Northwestern University Comprehensive Metabolic Core, and the Northwestern University Center for Advanced Microscopy (RRID: SCR_020996), supported by NCI CCSG P30 CA060553 awarded to the Robert H Lurie Comprehensive Cancer Center. This research was supported by the US Public Health Service NS122908 and NS124802, the US Department of Defense W81XWH-21-1-0236, and the Heather Koster Family Charitable Fund (RG Kalb); the NIH T32HL007909 and the Northwestern University Center for Sleep and Circadian Biology (SB Tam); and the Les Turner ALS Foundation (RG Kalb, E Kiskinis, SB Tam, M Wright, J Mojsilovic-Petrovic, and EM Baker). E Kiskinis is a New York Stem Cell Foundation Robertson Investigator.

### Author Contributions

SB Tam: conceptualization, resources, data curation, formal analysis, funding acquisition, validation, investigation, visualization, methodology, and writing—original draft, review, and editing.
NJ Waldeck: data curation, formal analysis, and visualization.
M Wright: data curation, formal analysis, and visualization.
J Mojsilovic-Petrovic: investigation.
EM Baker: investigation.
E Kiskinis: conceptualization, data curation, supervision, and methodology.
J Bass: conceptualization, data curation, supervision, methodology, and writing—review and editing.
RG Kalb: conceptualization, data curation, supervision, funding acquisition, methodology, and writing—review and editing.

### Conflict of Interest Statement

The authors declare that they have no conflict of interest.

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
