## [Reviewer comments · Life Science Alliance]

A role for the cholinergic neuron circadian clock in RNA metabolism and mediating neurodegeneration

Sharon B Tam, Nathan J Waldeck, Matthew Wright, Jelena Mojsilovic-Petrovic, Erin M Baker, Evangelos Kiskinis, Joseph Bass and Robert G Kalb

DOI: <https://doi.org/10.26508/lsa.202503508>

Corresponding author(s): Dr. Robert G Kalb (Northwestern University)

Review Timeline:

Submission Date:	2025-09-12
Editorial Decision:	2025-09-16
Revision Received:	2025-10-15
Editorial Decision:	2025-11-21
Revision Received:	2025-12-01
Accepted:	2025-12-03

Scientific Editor: Tim Fessenden

Transaction Report:

Please note that the manuscript was previously reviewed at another journal and the reports were taken into account in the decision-making process at *Life Science Alliance*. Since the original reviews are not subject to Life Science Alliance's transparent review process policy, the reports and author response cannot be published.

September 16, 2025

Re: Life Science Alliance manuscript #LSA-2025-03508-T

Robert G Kalb
Northwestern University Feinberg School of Medicine

Dear Dr. Kalb,

Thank you for submitting your manuscript entitled "Cholinergic neuron circadian clock mediates RNA-binding protein function and contributes to ALS disease phenotypes" to Life Science Alliance. In accordance with our offer of consideration conveyed by another journal, we invite you to submit a revised manuscript addressing key reviewer concerns.

Should you have any concerns about carrying out any of the above revisions, do please contact me and I'll be happy to discuss.

Please note that papers are generally considered through only one revision cycle, so strong support from the referees on the revised version is needed for acceptance. When submitting the revision, please include a letter addressing the reviewers' comments point by point and referencing our offer and guidance on revision.

Thank you for this interesting contribution to Life Science Alliance. We are looking forward to receiving your revised manuscript.

Sincerely,

- A letter addressing the reviewers' comments point by point.
- An editable version of the final text (.DOC or .DOCX) is needed for copyediting (no PDFs).
- High-resolution figure, supplementary figure and video files uploaded as individual files: See our detailed guidelines for preparing your production-ready images, <https://www.life-science-alliance.org/authors>
- Summary blurb (enter in submission system): A short text summarizing in a single sentence the study (max. 200 characters including spaces). This text is used in conjunction with the titles of papers, hence should be informative and complementary to the title and running title. It should describe the context and significance of the findings for a general readership; it should be written in the present tense and refer to the work in the third person. Author names should not be mentioned.
- By submitting a revision, you attest that you are aware of our payment policies found here: <https://www.life-science-alliance.org/copyright-license-fee>

B. MANUSCRIPT ORGANIZATION AND FORMATTING:

*****IMPORTANT:** It is Life Science Alliance policy that if requested, original data images must be made available. Failure to provide original images upon request will result in unavoidable delays in publication. Please ensure that you have access to all original microscopy and blot data images before submitting your revision. *******

November 21, 2025

RE: Life Science Alliance Manuscript #LSA-2025-03508-TR

Dr. Robert G Kalb
Northwestern University
Neurology
303 E Chicago Ave
Ward 13-270
Chicago, IL 60611

Dear Dr. Kalb,

Thank you for submitting your revised manuscript entitled "A role for the cholinergic neuron circadian clock in RNA metabolism and mediating neurodegeneration". We would be happy to publish your paper in Life Science Alliance pending those changes as well as final revisions necessary to meet our formatting guidelines.

- Please be sure that the authorship listing and order is correct.
- Please add ORCID ID for the corresponding author - you should have received instructions on how to do so.
- Please add the X and Bluesky handles of your host institute/organization, as well as your own and/or one of the authors, in our system.
- Please add your main, supplementary figure, and table legends to the main manuscript text after the references section.
- The contributions selected for Evangelos Kiskinis and Joseph Bass do not qualify them for authorship. Please either update the contributions in our system and in the Author Contributions section of the manuscript, or let us know if the authors need to be removed (and added potentially to the acknowledgment section).
- While the current title is appropriate, we suggest that you consider a slight change for brevity and improved readability:
"A role for the cholinergic neuron circadian clock in RNA metabolism and neurodegeneration"
- Please include molecular weight markers for the blots in Figure 2B.

LSA now encourages authors to provide a 30-60 second video where the study is briefly explained. We will use these videos on social media to promote the published paper and the presenting author (for examples, see <https://docs.google.com/document/d/1-UWCfbE4pGcDdcgzcmiuJl2XMBJnxKYeqRvLLrLS08s/edit?usp=sharing>). Corresponding or first-authors are welcome to submit the video. Please submit only one video per manuscript. The video can be emailed to contact@life-science-alliance.org

To upload the final version of your manuscript, please log in to your account: <https://lsa.msubmit.net/cgi-bin/main.plex>

A. FINAL FILES:

B. MANUSCRIPT ORGANIZATION AND FORMATTING:

Thank you for your attention to these final processing requirements. Please revise and format the manuscript and upload materials as soon as you are able.

Sincerely,

December 3, 2025

RE: Life Science Alliance Manuscript #LSA-2025-03508-TRR

Dr. Robert G Kalb
Northwestern University
Neurology
303 E Chicago Ave
Ward 13-270
Chicago, IL 60611

Dear Dr. Kalb,

Thank you for submitting your Research Article entitled "A role for the cholinergic neuron circadian clock in RNA metabolism and mediating neurodegeneration". It is a pleasure to let you know that your manuscript is now accepted for publication in Life Science Alliance. Congratulations on this interesting work and thank you for taking up our transfer offer following review at another journal. It was a pleasure working with you on this manuscript.

DISTRIBUTION OF MATERIALS:

Again, congratulations on a very nice paper. I hope you found the review process to be constructive and are pleased with how the manuscript was handled editorially. We look forward to future exciting submissions from your lab.

Sincerely,
